# Quantization of Large Language Models with an Overdetermined Basis

**Daniil Merkulov**[1,2,*]   **Daria Cherniuk**[1,*]   **Alexander Rudikov**[1,3]   **Ivan Oseledets**[1,3,4]   **Ekaterina Muravleva**[1,5]

**Aleksandr Mikhalev**[1]                           **Boris Kashin**[3,6]

[1]Skolkovo Institute of Science and Technology, Moscow, Russia ,
[2]Moscow Institute of Physics and Technology, Moscow, Russia ,
[3]Steklov Mathematical Institute of Russian Academy of Sciences, Moscow, Russia ,
[4]Artificial Intelligence Research Institute, Moscow, Russia ,
[5]Sberbank PJSC, Vavilova st., 19, 117312, Moscow, Russia
[6]M. V. Lomonosov Moscow State University, Moscow, Russia ,
[*]*Equal contribution*

## Abstract

In this paper, we introduce an algorithm for data quantization based on the principles of Kashin representation. This approach hinges on decomposing any given vector, matrix, or tensor into two factors. The first factor maintains a small infinity norm, while the second exhibits a similarly constrained norm when multiplied by an orthogonal matrix. Surprisingly, the entries of factors after decomposition are well-concentrated around several peaks, which allows us to efficiently replace them with corresponding centroids for quantization purposes. We study the theoretical properties of the proposed approach and rigorously evaluate our compression algorithm in the context of next-word prediction tasks and on a set of downstream tasks for text classification. Our findings demonstrate that Kashin Quantization achieves competitive or superior quality in model performance while ensuring data compression, marking a significant advancement in the field of data quantization.

## 1 INTRODUCTION

The realm of data processing and machine learning is increasingly confronting challenges linked to the efficient storage and management of large volumes of data. Traditional methods frequently result in elevated storage costs and computational inefficiencies, particularly when handling modern Large Language Models (LLMs). The availability of open source LLMs, such as OPT [Zhang et al., 2022], Llama 1 and 2 [Touvron et al., 2023a,b], Falcon [Almazrouei et al., 2023] stimulates the need to run them efficiently on local hardware.

In response, we introduce *Kashin Quantization*, a novel approach for data quantization that extends beyond the conventional use of the Kashin representation algorithm. We studied the algorithm for constructing Kashin representations [Kashin and Romskii, 2023], proposed a fast matrix version of the algorithm, and suggested an approach to use it for the quantization of large models.

The method re-imagines data representation by breaking down any data structure into two key factors. The first factor is optimized for a minimal infinity norm, while the second maintains a small infinity norm when post-multiplied by an orthogonal matrix. This strategy is not merely theoretical; its linear convergence makes it both effective and rapid in practical applications.

Addressing the challenge of working with resource-intensive orthogonal matrices, Kashin Quantization integrates structured matrix types, such as Householder, DCT, and Butterfly matrices. These matrices enable fast matvec operations, boosting memory efficiency and accelerating computation, thereby elevating the overall effectiveness of the algorithm.

A pivotal aspect of Kashin Quantization is its proficiency in concentrating data values around defined peaks (Figure 6), substantially reducing the need for expansive storage. By mapping these values to centroids of two-dimensional distributions (one dimension for each factor), it remarkably reduces data representation size, compressing traditional 32-bit floats to low-bit values. This technique is notably advantageous in large-scale machine learning models, where data size and efficiency are paramount.

Expanding its utility, Kashin Quantization is not only pertinent for quantizing Large Neural Networks but also for compressing exchange information, like gradients in Federated Learning [Safaryan et al., 2022] and distributed computations. This extension illustrates the versatility of our approach in diverse machine learning environments.

Furthermore, we enhance Kashin Quantization by incorporating a matrix reformulation of the Kashin algorithm, which substantially accelerates computations. This advancement provides a more efficient methodology for handling large-scale data and complex computations in neural net-

works.

To validate our approach's efficacy, we apply Kashin Quantization to several common benchmarks in natural language processing. Employing language models like OPT, Bert, and RoBerta we demonstrate that our method leads to competitive or superior predictive quality on both next token prediction and text classification tasks while maintaining the level of quantization. This balance of efficiency and performance highlights Kashin Quantization's potential to transform data quantization practices in machine learning and allied fields.

This paper makes the following contributions:

1. We introduce the Matrix Decomposition Kashin Algorithm. The vanilla algorithm for building Kashin representation requires vectorization of input data and building an orthogonal matrix of possibly huge dimensions. For an $m \times n$ matrix to decompose, we need a matrix of $(mn)^2$ elements. Instead, we propose to use a Kronecker-factored matrix, which leads to the memory footprint of $m^2 + n^2$. We benchmark the proposed approach and study its limitations.

2. We explore the theoretical properties of the convergence rate of the algorithm we proposed. We observed, that the specific choice of basis vectors affects the convergence. We establish the connection between the convergence rate of the Vector Decomposition Kashin Algorithm and the Kolmogorov width, corresponding to the specific choice of basis.

3. We propose the Kashin Quantization algorithm for neural network weights quantization, that utilizes the properties of the Kashin representations. We benchmark the proposed approach empirically on the family of OPT models on the next token prediction problem and compare perplexity with the baselines, demonstrating the validity of the proposed approach. Additionally, we conduct experiments on the GLUE benchmark, quantizing Bert and RoBerta models finetuned on several downstream tasks and achieving superior results compared to common quantization methods.

## 2 METHOD

**Representing the vector in overdetermined basis.** Suppose we have a vector $x \in \mathbb{R}^n$. Its elements may have different magnitudes (large 'spread') and that can prohibit efficient quantization (i.e., low-bit representation) of the elements. We aim to transform this vector into another vector, which can be represented with fewer number of bits. First, we introduce an overdetermined basis of size $2n$: instead of $n$ coefficients, we will represent the vector with $m = 2n$ coefficients, but with fewer bits per element. This basis will consist of the original standard basis and the basis given by

the columns of the orthogonal matrix $Q$. In vector notation, this is equivalent to representing $x$ as a sum

$$x \approx u + Qv, \tag{1}$$

which admits good low-bit approximation (we will discuss this in detail in Section 4). Fortunately, based on the results of Kashin [1977] and Kashin and Romskii [2023] this representation can be computed efficiently for an arbitrary $Q$ except for the set of small measure:

**Theorem 1** (Kashin and Romskii [2023]). *For $\forall x \in B_2^N = \{x \in \mathbb{R}^N : \|x\|_2 \leq 1\}$ a greedy algorithm in $k$ steps builds vectors $u_k$ and $v_k$ such that*

$$\|x - u_k - v_k\|_2 \leq \eta^k$$

$$\|u_k\|_\infty \leq \frac{c}{\sqrt{N}}, \quad \|Qv_k\|_\infty \leq \frac{c}{\sqrt{N}}$$

*where $c > 0$ is an absolute constant, $Q \in O(N)$, where $O(N)$ is a set of orthogonal matrices in $\mathbb{R}^N$ with Haar measure $\mu_H$.*

**Computing the representation using a greedy approach.** Without loss of generality, we assume that the vector $x$ has a unit Euclidean norm $\|x\|_2 = 1$ (otherwise it can be scaled). Given $x$, we can find the closest vector $\hat{x}$ with elements $\hat{x}_i \in \{-1, 1\}$ as $\hat{x}_i = \text{sign}(x_i)$, note, that $\|\hat{x}\|_2 = \sqrt{n}$. The chosen vector serves as the direction for the first vector $u$ in the relation (1). Which means, that the projection of $x$ onto $\hat{x}$ is

$$\pi_x(\hat{x}) = \lambda \hat{x} = \|x\|_2 \frac{\langle x, \hat{x} \rangle}{\|x\|_2 \|\hat{x}\|_2} \frac{\hat{x}}{\|\hat{x}\|_2} = \frac{\|x\|_1}{n} \hat{x}.$$

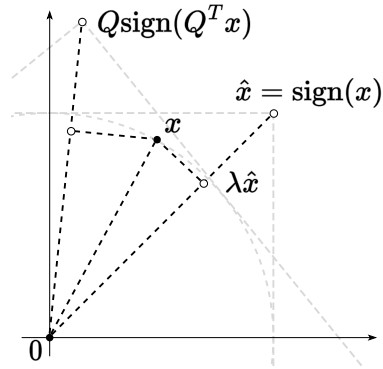

Figure 1: We select the vector to project on in a greedy manner.

and since three vectors $x$, $\lambda \hat{x}$ and $x - \lambda \hat{x}$ will always form an orthogonal triangle by design the distance is given as

$$\|x - \lambda \hat{x}\|_2^2 = \|x\|_2^2 - \frac{\|x\|_1^2}{n^2}\|\hat{x}\|_2^2 = \|x\|_2^2 - \frac{\|x\|_1^2}{n}. \tag{2}$$

**Algorithm 1** Vector Decomposition Kashin Algorithm

**Input:** Vector $x \in \mathbb{R}^n$, Orthogonal matrix $Q$, Tolerance $\varepsilon > 0$

**Output:** Vectors $u, \hat{v} \in \mathbb{R}^n$ such that $x \approx u + \hat{v} = u + Qv$, and both $u$ and $v$ have small infinity norm.

Initialize $u \leftarrow 0^n, \hat{v} \leftarrow 0^n$

Define projection $\pi_x(y) := \dfrac{x^\top y}{\|y\|_2^2} \cdot y$

**while** $\|x - u - \hat{v}\| \geq \varepsilon$ **do**
  **if** $\|x\|_1 > \|Q^T x\|_1$ **then**
    $\pi \leftarrow \pi_x(\text{Sign}(x))$
    $u \leftarrow u + \pi$
  **else**
    $\pi \leftarrow \pi_x(Q\text{Sign}(Q^T x))$
    $\hat{v} \leftarrow \hat{v} + \pi$
  **end if**
  $x \leftarrow x - \pi$
**end while**
**Return:** $x, u, \hat{v}$

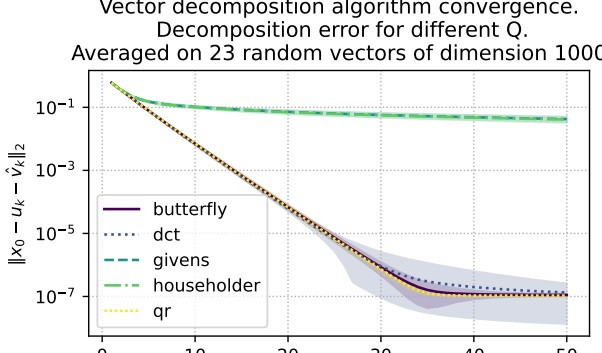

Figure 2: Convergence criterion ($\|x_0 - u_k - \hat{v}_k\|_2$) for randomly generated vector $x$ of length 1000. The semi-transparent area shows the confidence interval.

The second component $Qv$ from (1) will be formed similarly. Note, that multiplying the vector $v$ with $\pm 1$ components by matrix $Q$ is equivalent to the rotation of the whole space with the matrix $Q$, which is also the same as the backward rotation of all vectors except $v$ with the matrix $Q^T$. Indeed, from (1) $Q^T x \approx Q^T u + v$. Thus, the closest vector to $x$ of the form $Qv$ with $v_i \in \{-1, 1\}$ is given as

$$\hat{x} = \lambda Q \text{sign}(Q^T x),$$

and the residual given by the Pythagorean theorem again is

$$\|x - \hat{x}\|_2^2 = \|x\|_2^2 - \frac{\|Q^T x\|_1^2}{n}. \tag{3}$$

In the greedy algorithm we start from $x_0 = x$, then for $k = 0, \ldots$ we select a vector of form $\lambda \hat{x_k}$ or $\lambda Q \hat{x_k}$ (where $\hat{x_k}$ has elements $\pm 1$) that is the closest to the vector $x_k$, subtract the projection $\pi_{x_k}(\text{sign}(x_k))$ or $\pi_{x_k}(\text{sign}(Q^T x_k)))$ from it and repeat the procedure until convergence. Thus, we decompose the initial vector decreasing the norm of $x_k$ each iteration and increasing the norm of factors $u_k$ and $\hat{v}_k$. The idea is summarized in Algorithm 1.

**Convergence of the algorithm** The vector algorithm is guaranteed to converge for arbitrary orthogonal matrix $Q$ except for a set of small measure [Kashin, 1977]. Our experiments demonstrate, that the convergence speed varies when choosing from the classes of structured orthogonal matrices. The demonstration is presented in Figure 2, where we took 5 different orthogonal matrices $Q$ and generated 23 random vectors of the form $x \in \mathbb{R}^{1000}$. Householder reflection and Givens rotation perform poorly, whereas dct, random orthogonal and butterfly matrices perform on par.

Differences in convergence raise a natural question about the convergence properties of Algorithm 1. On each iteration we have a vector as an input $x$, from which we subtract a projection, resulting in a vector $x - \hat{x}$. Thus, we consider the difference between them as a decreasing factor on the iteration and any constructive approach to bound its norm with respect to the matrix $Q$ properties is of interest.

Suppose on $k$-th iteration we have $x_k$ and the squared norm after a single iteration of the algorithm $\|x_{k+1}\|_2^2$ could be either $\|x\|_2^2 - \frac{\|x\|_1^2}{n}$ or $\|x\|_2^2 - \frac{\|Q^T x\|_1^2}{n}$ from (2) and (3). Note also, that the choice is made in favor of maximum 1-norm $\max(\|x\|_1, \|Q^T x\|_1)$. Overall, we decrease the residual norm at each iteration, which ensures the convergence of the algorithm. One can consider starting iteration, where $\|x_0\|_2 = \|x\|_2 = 1$, which leads to the

$$\|x - \hat{x}\|^2 \leq 1 - \frac{1}{n}\left(\max(\|x\|_1, \|Q^T x\|_1)\right)^2 \tag{4}$$

Clearly, the specific choice of the matrix $Q$ affects the quantity $\max(\|x\|_1, \|Q^T x\|_1)$. Moreover, this simple estimation immediately poses another interesting question. How does the choice of an input vector $x$ affect the convergence? The answer could be obtained from studying the following optimization problem, which is clearly related to the Kolmogorov width Temlyakov [1998] estimation:

$$\min_{\|x\|_2 = 1} \max(\|x\|_1, \|Q^T x\|_1) \tag{5}$$

The problem (5) determines the worst-case convergence rate for a specific matrix $Q$. To address the problem we considered several classes of orthogonal matrices $Q$ (see more details in Section 3) and performed a search across all eigenvectors of the matrix $Q^T$ ($\|Q^T x\|_1 = \|\lambda x\|_1$). The results are presented in Table 1.

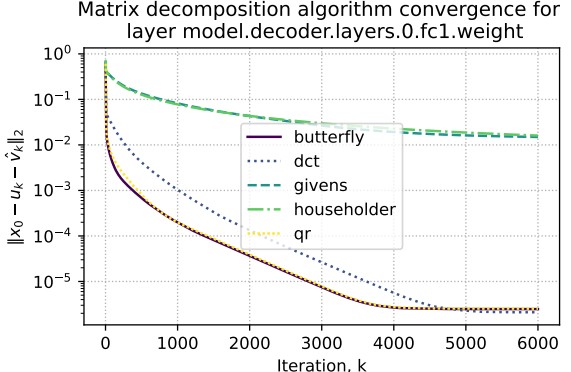 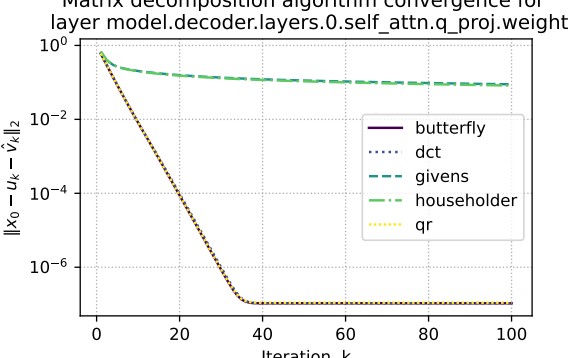

Figure 3: Convergence criterion ($\|x - \hat{x}\|_2$) of the matrix decomposition Kashin algorithm for different layers of OPT-125m model. Left: The algorithm with the DCT matrix converges very slowly after several iterations. Right: The algorithm with the DCT matrix converges well for another layer.

**Matrix version of the algorithm.** As we delved into the nuances of matrix decomposition, leveraging the pioneering Kashin vector decomposition algorithm 1 initially seemed a straightforward extension: vectorize the matrices and proceed with the decomposition. Yet, this simplistic approach soon revealed its limitations, especially when faced with matrices of even moderate size, such as the $768 \times 3072$ matrices encountered in the fully connected layers of the OPT-125m model. The challenge of managing large vectors soon made it clear that we needed a new approach. This led to the development of the Matrix Decomposition Kashin Algorithm (Algorithm 2), an improved version of the earlier method, specifically created to tackle the difficulties involved in breaking down matrices.

In the naive implementation to decompose a matrix of shape $m \times n$ one needs to store matrix $Q$ of shape $mn \times mn$, while the matrix version leads to memory reduction since we need to store $m \times m + n \times n$ for $Q_1$ and $Q_2$. The genesis of the proposed Algorithm 2 follows from the simple observation. Let's assume our orthogonal matrix $Q$ can be represented as a Kronecker product of two smaller orthogonal matrices $Q_1^T$ and $Q_2^T$:

$$Q = Q_2^T \otimes Q_1^T \tag{6}$$

From the properties of Kronecker product, we get:

$$(Q_2^T \otimes Q_1^T)\text{vec}(X) = \text{vec}(Q_1^T X Q_2) \tag{7}$$

However, by assuming the special structure of $Q$ we might lose the robustness guaranteed for random orthogonal matrices (as proven in [Kashin, 1977], [Lyubarskii and Vershynin, 2010]). Indeed, as shown in [Guedon et al., 2008], some discrete orthonormal systems produce slightly worse bounds.

Figure 4 illustrates that the time needed for the same compu-

---

**Algorithm 2** Matrix Decomposition Kashin Algorithm

**Input:** Matrix $X \in \mathbb{R}^{m \times n}$, Orthogonal matrices $Q_1, Q_2$, Tolerance $\varepsilon > 0$
**Output:** Matrices $U, \hat{V} \in \mathbb{R}^{m \times n}$, such that $X \approx U + \hat{V} = U + Q_1 V Q_2^T$ and both $\text{Vec}(U)$ and $\text{Vec}(V)$ have small infinity norm.
Initialize $U \leftarrow 0^{m \times n}, \hat{V} \leftarrow 0^{m \times n}$
Define projection $\pi_X(Y) := \dfrac{\text{Vec}(X)^\top \text{Vec}(Y)}{\|\text{Vec}(Y)\|_2^2} \cdot Y$
**while** $\|X - U - \hat{V}\| \geq \varepsilon$ **do**
    $Y \leftarrow Q_1^T X Q_2$
    **if** $\|\text{Vec}(X)\|_1 > \|\text{Vec}(Y)\|_1$ **then**
        $\pi \leftarrow \pi_X(\text{Sign}(X))$
        $U \leftarrow U + \pi$
    **else**
        $\pi \leftarrow \pi_X(Q_1 \text{Sign}(Y) Q_2^\top)$
        $\hat{V} \leftarrow \hat{V} + \pi$
    **end if**
    $X \leftarrow X - \pi$
**end while**
**Return:** $X, U, \hat{V}$

---

tations is notably less for the matrix version of the algorithm than for the vector one.

# 3 HOW ORTHOGONAL MATRIX $Q$ AFFECTS THE ALGORITHM

The choice of orthogonal matrix $Q$ in algorithms 1 and 2 can significantly impact convergence and execution time. Figure 3 demonstrates the different convergence properties of the algorithm for different input matrices.

In this work, we have tried several classes of orthogonal matrices:

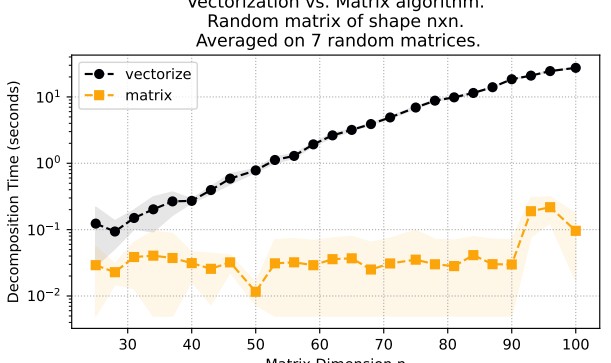

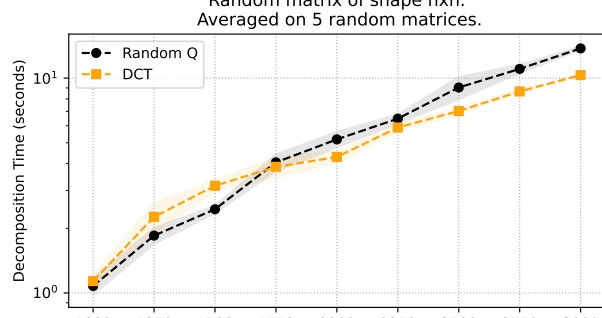

Figure 4: Left: Matrix Decomposition Kashin Algorithm 2 performs significantly faster for matrices, than vectorization and applying Vector Decomposition Kashin Algorithm 1. Right: Time comparison of different orthogonal matrices for Matrix Decomposition Kashin Algorithm

## 3.1 RANDOM ORTHOGONAL MATRIX

Random orthogonal matrix $Q$ is obtained from the QR decomposition of a matrix generated from a normal distribution.

## 3.2 HOUSEHOLDER REFLECTION

The Householder matrix is an orthogonal matrix that describes a reflection across a hyperplane orthogonal to some unit vector $y$:

$$Q = I - 2yy^* \tag{8}$$

## 3.3 DISCRETE COSINE TRANSFORM

*Discrete Cosine Transform* (DCT) is an orthogonal linear mapping similar to the discrete Fourier transform (DFT), but using only the cosine functions. We chose the most commonly used form, DCT-II:

$$Q[i,j] = \begin{cases} \frac{1}{\sqrt{n}}, \ j = 0 \\ \sqrt{\frac{2}{n}} \cos\left(\frac{\pi(2i+1)j}{2n}\right) \end{cases} \tag{9}$$

## 3.4 BUTTERFLY MATRICES

*Butterfly matrices* are a class of structured orthogonal matrices that allow fast vector-by-matrix multiplication.

A *butterfly matrix* $Q$ of size $n = 2^m$ can be represented as a product of $m$ block diagonal matrices (called *butterfly factor matrices*). We denote each butterfly factor matrix as $B_k$, where $k$ indicates block size:

$$\begin{aligned} Q &= B_n B_{\frac{n}{2}} \dots B_2 \\ B_k &= \mathrm{diag}(F_1, F_2, \dots, F_{\frac{n}{k}}) \end{aligned} \tag{10}$$

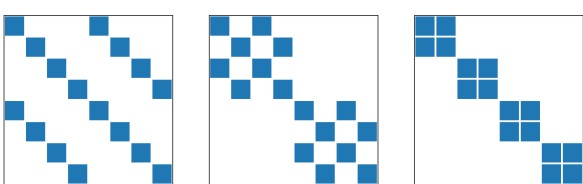

Figure 5: Butterfly factor matrices from left to right: $B_8$, $B_4$, $B_2$

Each block $F_i$ (a *butterfly factor*) of size $k$ has a form $\begin{bmatrix} D_1 & D_2 \\ D_3 & D_4 \end{bmatrix}$ where $D_i$ is a $\frac{k}{2} \times \frac{k}{2}$ diagonal matrix. Figure 5 shows an example of a butterfly matrix of size 16.

Matrices like Householder Reflection (3.2), Discrete Cosine Transform (3.3), and Butterfly matrices (3.4) support faster matvec multiplications (faster than $O(n^2)$) due to their special structure. For example, Householder transformation boils down to a simple operation with two vectors:

$$(I - 2yy^*)x = x - 2\langle y, x \rangle y \tag{11}$$

while DCT and Butterfly matrices allow for $O(n \log n)$ complexity of matrix-vector multiplication.

## 3.5 CONVERGENCE PROPERTIES

However, an obvious problem arises for Householder Reflexion considering the convergence bound (4): when $\|x\|_1$ is small (for example, $x$ is very sparse) and the dimension is high, vectors $y$ and $x$ might be orthogonal, thus $Q^T x$ will be equal to $x$ ($x$ will lie on the reflection plane and will be the eigenvector of $Q^T$). This keeps $\max(\|x\|_1, \|Q^T x\|_1)$ small and, consequently, the convergence bound - high.

We try to estimate potential minimums of $\max(\|x\|_1, \|Q^T x\|_1)$ expression in Table 1 by taking

Table 1: Estimation for $x = \text{Re}[y]$, $y \in \text{eig}(Q)$. Mean over 100 generated Q matrices (except for DCT). Matrix size is $500 \times 500$. All results are divided by $\sqrt{n}$.

| $Q$ | $\min_{\|x\|_2=1} \max\{\|x\|_1, \|Q^T x\|_1\}$ |
|---|---|
| RANDOM | **0.548** |
| HOUSEHOLDER | 0.076 |
| DCT | 0.330 |
| BUTTERFLY | **0.547** |

the real part of eigenvalues (only the Householder matrix is symmetric and has real eigenvectors) of each orthogonal matrix. These results allow us to hypothesize that *qr* and *butterfly* matrices in the general case should converge best. These results coincide with the empirical convergence of the algorithm for randomly generated vectors (Figure 2).

## 4 QUANTIZATION APPROACH

As soon as we have the decomposition $X = U + Q_1 V Q_2^T$, where $u$ and $v$ have small infinity norm, we may hope, that the entries distribution of these vectors may be quantized well. The idea behind the quantization approach is fairly simple. We replace the values in the factors with the nearest centroids values, calculated with the clustering procedure. For illustration, we consider matrix $X \in \mathbb{R}^{500 \times 200}$ with entries from standard distribution and apply Matrix Decomposition Kashin Algorithm 2 to it. Figure 6 clearly illustrates that values of factors $U$ and $V$ are well quantized.

The red lines and dots on the graph represent the centroids of corresponding clusters of values. The number of clusters is defined by the number of bits for quantization. Figure 6 contains 4 clusters per factor, which means, that we need to store the values of $16 = 2^4$ points or 4 bits. From the same figure, one can conclude, that 16 clusters per factor leads to the $16 \times 16 = 256 = 2^8$ possible pairs of clusters.

To measure the quality of the proposed quantization procedure we decided to consider weights of OPT models [Zhang et al., 2022] and decompose all layers except the embedding layer in the model in a similar manner. So, we applied Algorithm 2 for each weight matrix $X_i$ and recieved $U_i$, $\hat{V}_i$ for layer $i$. After that, we applied the clustering procedure to the set of 2-dimensional vectors obtained from stacking $U_i$ values and $V_i = Q_1^T \hat{V}_i Q_2$ values with a selected number of clusters (32 for 5 bits, 64 for 6 bits, etc.). After obtaining centroids, we replace each entry in the matrices $U_i$, $V_i$ with its closest centroid value. As a result, we have quantized factors $U_i^q$ and $V_i^q$ with relatively small infinity error. And all we have to do is to replace the old forward pass with matrix $X_i(\cdot)$ with the new forward pass $\left[U_i^q + Q_1 V_i^q Q_2^T\right](\cdot)$. It is important to highlight, that we do not need to store the

Table 2: Perplexity for OPT family of models with different quantization schemes. Part of the transformer layers are quantized with Kashin M-bit quantization (those, for which Kashin decomposition has converged) and others are quantized with N-bit. The "Quantized Layers" column specifies how many layers were quantized with each method (first fraction for N-bit and second fraction for kM-bit).

| QUANTIZATION | PERPLEXITY | QUANTIZED LAYERS |
|---|---|---|
| **OPT-125M** | | |
| FP32 | 42.01 | |
| 8-BIT | 42.19 | 72/72 |
| 8-BIT + K6-BIT | 43.38 | 15/72 + 57/72 |
| 8-BIT + K5-BIT | 45.74 | 15/72 + 57/72 |
| 4-BIT | 48.92 | 72/72 |
| 4-BIT + K6-BIT | 44.88 | 15/72 + 57/72 |
| 4-BIT + K5-BIT | 47.25 | 15/72 + 57/72 |
| **OPT-350M** | | |
| FP32 | 36.64 | |
| 8-BIT | 36.77 | 146/146 |
| 8-BIT + K6-BIT | 37.59 | 25/146 + 121/146 |
| 8-BIT + K5-BIT | 40.01 | 25/146 + 121/146 |
| 4-BIT | 44.46 | 146/146 |
| 4-BIT + K6-BIT | 37.61 | 25/146 + 121/146 |
| 4-BIT + K5-BIT | 40.05 | 25/146 + 121/146 |

matrices $Q_1$ and $Q_2$, their action is replaced with effective DCT or Butterfly matvec procedures.

However, we observed, that not all layers could be compressed equally well with Algorithm 2. This is clearly distinguished either visually (see Figure 7) or from monitoring convergence speed. The reasons behind such behavior of the method are yet to be described. For practical reasons, we decided not to decompose such layers. The results are presented in Table 2.

## 5 EXPERIMENTS

To prove the efficacy of our method, we use the Open Pretrained Transformers (OPT [Zhang et al., 2022]) language models and the LAMBADA dataset [Paperno et al., 2016].

First, we try to compare Kashin Decomposition with different orthogonal matrices for several layers of the opt-125m language model. Unlike randomly generated matrices, the convergence of Kashin decomposition for network weights can be quite unpredictable. Figure 3 depicts the value of the convergence criterion for two different layers of the first transformer block. For the fc1 layer, the algorithm fails to

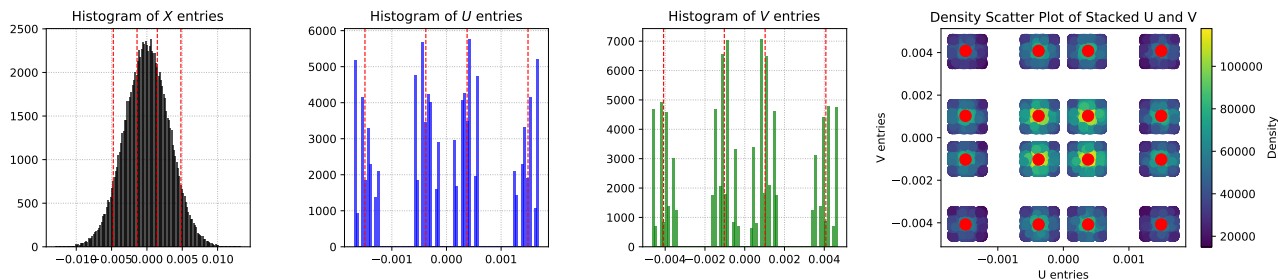

Figure 6: Factors $U$ and $V$ are well quantized after computing Kashin decomposition.

Table 3: GLUE

| QUANTIZATION | COLA | SST-2 | QQP | QNLI | MNLI | RTE | STS-B | MRPC | WNLI |
|---|---|---|---|---|---|---|---|---|---|
| **ROBERTA-BASE** | | | | | | | | | |
| FP32 | 59.06 | 93.8 | 91.24/88.36 | 92.62 | 88.1/87.43 | 67.87 | 89.65/89.49 | 87.75/91.2 | 56.34 |
| UNIFORM 4BIT | 0.0 | 49.08 | 36.82/53.82 | 49.46 | 31.82/31.82 | 52.71 | 10.08/9.37 | 68.38/81.22 | 56.34 |
| KMEANS 4BIT | 46.97 | **92.77** | 88.77/**87.39** | 89.27 | 83.98/82.85 | 55.23 | 79.65/80.47 | 71.32/75.77 | 56.34 |
| KASHIN 4BIT (OURS) | **52.09** | 90.37 | **89.53**/86.73 | **91.14** | **86.41/85.51** | **60.28** | **87.29/87.26** | **83.08/87.10** | 56.34 |
| **BERT-BASE** | | | | | | | | | |
| FP32 | 59.31 | 91.74 | 90.66/87.39 | 90.74 | 83.96/84.24 | 64.98 | 88.94/88.77 | 84.31/88.81 | 42.25 |
| UNIFORM 4BIT | 1.24 | 49.66 | 38.09/52.75 | 49.22 | 32.24/33.34 | 50.18 | -0.21/-0.25 | 63.97/76.02 | 49.3 |
| KMEANS 4BIT | 54.43 | 91.51 | 88.87/85.33 | 88.01 | 78.63/78.76 | 55.23 | 85.06/85.0 | 34.55/8.87 | **54.92** |
| KASHIN 4BIT (OURS) | **59.65** | **91.63** | **90.28/87.33** | **90.06** | **83.88/84.01** | **63.53** | **88.76/88.56** | **84.80/89.31** | 42.25 |

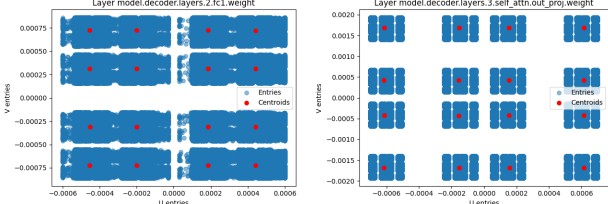

Figure 7: Left: Slow convergence of the Matrix Decomposition Kashin algorithm for a particular weight matrix from OPT-125m leads to poor concentration of values and high quantization error. Right: another layer weight matrix could be easily decomposed and quantized.

converge in the maximum number of steps and *dct* converges slower than *butterfly* and *qr*. For the q_proj layer, *dct* performs on par with *qr* and slightly better than *butterfly*. Kashin Decomposition with *householder reflection* performs poorly in both cases. In all further experiments, we use either *qr* or *butterfly* orthogonal matrices.

Next, we apply Kashin Quantization and measure LLM's perplexity. As seen in Figures 3 and 7, Kashin Decomposi-

tion doesn't always converge for LLM weights, which leads to high quantization error. In our experiments with the OPT family of models, we encountered about 0.2 fractions of poorly converged layers. To compensate for these layers, we leave their weights with the default quantization scheme (8-bit or 4-bit versions of weights from the huggingface hub). Results can be seen in Table 2.

Kashin Quantization achieves a trade-off between low bit weights, inference speed, and LLM prediction quality. For example, full 4-bit quantization significantly raises the model's perplexity, whereas 4-bit + Kashin 6-bit quantization almost restores the quality of the 8-bit model. That, in conjunction with the ability to perform lower-bit operations and faster matrix multiplication (with the right orthogonal matrix choice, for example, butterfly matrices) can significantly reduce LLM's inference speed and memory requirements.

Additionally, we conduct experiments on text classification with a set of downstream tasks (GLUE benchmark). We have finetuned Bert and RoBerta models, measured their accuracy as a baseline, and then quantized linear layers in transformer blocks with three quantization methods: uniform, kmeans,

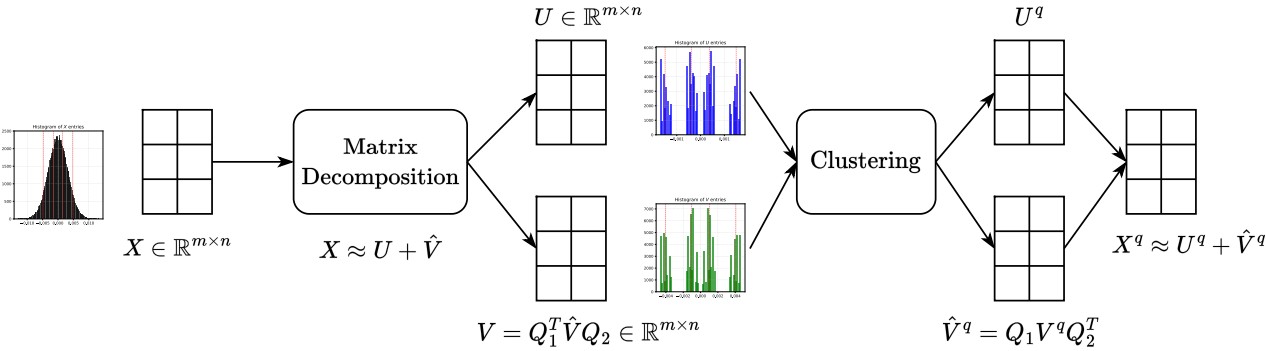

Figure 8: Scheme of post-training quantization approach.

and Kashin quantization (ours). Kashin quantization demonstrates superior quality on almost all GLUE datasets for both models (Figure 3).

# 6 RELATED WORK

Large Language Models (LLMs) nowadays are ubiquitous and used in all imaginable applications. However, their complexity heavily depends on several parameters. For example, the flagship of LLAMA-2 [Touvron et al., 2023a] family of models has over 70 billion parameters. This impedes model speed and sometimes makes training or even inference impossible due to a lack of GPU memory.

Quantization [Nagel et al., 2021] is a popular technique to reduce the model size and inference time. Naturally, it is often applied to LLMs. For instance, QLORA [Dettmers et al., 2023] paper used quantized weights in conjunction with Low-Rank Adapters to fine-tune language models on downstream tasks, and LLM.int8 [Dettmers et al., 2022] has introduced a new Int8 matrix multiplication procedure for feed-forward and attention projection layers in transformers. Still, the quantization of transformers often proves to be a tricky task, especially the quantization of activations. It happens due to the presence of huge outliers in activations that originate from the attention mechanism ([Bondarenko et al., 2021], [Bondarenko et al., 2023]).

Best accuracy results are usually obtained with quantization-aware training. Another approach, taken in [van Baalen et al., 2020], is to learn bit-width and quantization parameters simultaneously with model weights. However, both techniques require dataset and resources to train the model with an additional step of quantization on each weight update.

To avoid training expenses, methods that allow for no training/fine-tuning have been of particular interest in several papers: authors in [Eldad et al., 2019] scale the weights of consecutive layers which allows them to achieve smaller quality drop on a wide variety of networks;

AdaRound [Nagel et al., 2020] proves that rounding to the nearest node of the quantization grid is not the best strategy and propose to choose rounding by optimization process which requires only a few thousand samples of data, researches in [Hubara et al., 2020] further develop the idea of AdaRound, complementing it with integer programming to determine the best bit-width for each layer.

It is important to note that there are other ways to reduce memory footprint when using large models, such as automatic mixed precision [Micikevicius et al., 2017], operations approximation [Bershatsky et al., 2022, Novikov et al., 2023], and checkpointing [Chen et al., 2016, Gusak et al., 2022].

In our work we use *Kashin Representations*, introduced in [Kashin, 1977]. It's an expansion of vector $x$ in a redundant orthogonal basis which guarantees that the maximum among absolute values of coefficients (the infinity norm) is upper bounded. It is a source for representation in Eq. 1 and Algorithm 1. [Garnaev and Gluskin, 1984] discusses the redundancy level in Kashin representations and [Lyubarskii and Vershynin, 2010] investigates the applicability of Kashin representations to vector quantization using tight orthogonal frames that satisfy uncertainty principle.

# 7 CONCLUSION AND FURTHER RESEARCH

We have proposed a matrix case generalization of the Kashin Representation algorithm and applied it to the weights of transformer language models. We have compared different choices of orthogonal matrices and theorized on efficacy of each based on both empirical results and metrics in Eq. 5 estimation among eigenvectors of $Q$.

Kashin Quantization backed up by the conventional uniform quantization achieves a good compromise between model perplexity and memory requirements. Additionally, Kashin representation allows to use of faster matrix multiplication due to the choice of $Q$.

There is much yet to be determined about the properties of convergence of Algorithm 2. As noted in Section 5, the proposed decomposition for matrices of model weights doesn't always converge in a restricted number of iterations. The proper investigation of both failed matrix weights and special structure of orthonormal basis $Q$ is a question of further research.

We are looking forward to applying Kashin Decomposition to the quantization of activations. As shown in [Bondarenko et al., 2021] and [Bondarenko et al., 2023], quantization of transformer activations often suffers from huge outliers. Since Kashin Representation guarantees an upper bound on infinity norms (maximum absolute value) of decomposition factors, it can be assumed that Kashin Quantization should be particularly beneficial to LLM activations quantization.

## Acknowledgements

The work was supported by the Analytical center under the RF Government (subsidy agreement 000000D730321P5Q0002, Grant No. 70-2021-00145 02.11.2021)

B. S. Kashin's work on this article was supported by the grant No. 24-11-00114 from the Russian Science Foundation in M. V. Lomonosov Moscow State University .

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
