# OpenReview forum: "Quantization of Large Language Models with an Overdetermined Basis"
_auai.org/UAI/2024/Conference — UAI 2024 poster_

### Official Review · Reviewer_4SN6 · 2024-03-21

**Q2-1 Originality-Novelty:** 3
**Q2-2 Correctness-Technical Quality:** 3
**Q2-5 Clarity Of Writing:** 3

**Q1 Summary And Contributions:**

This paper introduces Kashin algorithms for vector/matrix decomposition, and applies them to the quantization of LLMs such as OPT. Theoretical analysis is provided for the proposed algorithm. Empirical results show that the proposed algorithm achieves competitive quality in model performance with compression.

**Q2-3 Extent To Which Claims Are Supported By Evidence:**

2: Fair: the main claims are somewhat supported by evidence (but the experimental evaluation may be weak, or does not match entirely with the claims, important baselines may be missing, proofs contain important ideas but lack rigor, algorithmic details are only discussed superficially, references are imprecise, assumptions are not sufficiently motivated or explicated, etc.).

**Q2-4 Reproducibility:**

2: Fair: key resources (e.g. proofs, code, data) are unavailable but key details (e.g. proof sketches, experimental setup) are sufficiently well-described for an expert to confidently reproduce the main results.

**Q3 Main Strengths:**

1. This paper introduces Kashin algorithms for vector/matrix decomposition, and applies them to the quantization of LLMs such as OPT.
2. Theoretical analysis is provided for the proposed algorithm.
3. Empirical results show that the proposed algorithm achieves competitive quality in model performance with compression.

**Q4 Main Weakness:**

1. The experiments are limited to OPT model and next-word prediction. Even if we stick to NLP tasks, there are other options of open-sourced models, including fine-tuned models which could be evaluated on some down-stream tasks. The current configuration of the experiments is not comprehensive enough, which makes the results less convincing.

2. In Table 2, it is shown that by replacing the default 8-bit quantization with the proposed Kashin quantization with lower bits, the perplexity also gets worse, which doesn't really seems to be an improvement to me.

**Q5 Detailed Comments To The Authors:**

1. For Table 2, could the authors show the total number of bits of the entire model, so that the readers could clearly get the results of compression ratio vs. perplexity? Although the columns here (quantization, quantized layers) shows the detailed configuration of the experiments, I cares more about the compression ratio here and don't want to calculate the overall compression ratio by myself when reading this table.

2. Note that for simple quantization, we could also do the compression with 5 or 6 or 7 bits (with fake/simulated quantization), and compare the results to the similar settings such as "8-bit + k6-bit" (kind of between 6bit and 7bit). Such baselines will show whether the proposed algorithm is really better than the default quantization methods with the same (or close enough) compression ratio.

3. In Table 2, I also strongly recommend to show the baseline without any compression.

**Q9 Complying With Reviewing Instructions:**

Yes

---

> ### Author Rebuttal · Authors · 2024-04-09
>
> Dear Reviewer, thanks for your feedback on the submission. We would like to carefully address the points you mentioned in weaknesses.
> * `The experiments are limited to OPT model and next-word prediction. Even if we stick to NLP tasks, there are other options of open-sourced models, including fine-tuned models which could be evaluated on some down-stream tasks. The current configuration of the experiments is not comprehensive enough, which makes the results less convincing.`
>
>     We have added [experiments](https://github.com/quantizeit/kashin/blob/main/figures/glue.jpeg?raw=true) with two more models (BERT and RoBERTa) on text-classification tasks (GLUE datasets) where we provide a comparison to uniform quantization and kmeans quantization using such metrics as accuracy, f1-score, Pearson/Spearman correlation, Matthews correlation.
>
> * `In Table 2, I also strongly recommend to show the baseline without any compression.`
>
>     As you suggested, we have included the perplexity of the model with fp32 weights in Table 2.
> * `In Table 2, it is shown that by replacing the default 8-bit quantization with the proposed Kashin quantization with lower bits, the perplexity also gets worse, which doesn't really seems to be an improvement to me.`
>
>     Indeed, however, the drop is not drastic for such a metric as perplexity. Furthermore, we consider our main result on opt models to be for the 4bit quantization of a larger opt-350m: 4bit + kashin6bit quantization almost recovers the perplexity of the 8bit model. Additionally, we realize that the comparison we provided for opt models may be not sufficient enough, so we have conducted [experiments](https://github.com/quantizeit/kashin/blob/main/figures/glue.jpeg?raw=true)  on Glue, we quantized all layers of BERT/RoBERTa models, not falling back onto other methods and still produced superior results on this benchmark.
> * `Although the columns here (quantization, quantized layers) shows the detailed configuration of the experiments, I cares more about the compression ratio here and don't want to calculate the overall compression ratio by myself when reading this table.`
>
>     We agree that more details are needed on the overall reduction rate in memory and inference speed. We would like to provide this information alongside the model performance quality results in the edited version of the paper.
> * `Note that for simple quantization, we could also do the compression with 5 or 6 or 7 bits (with fake/simulated quantization), and compare the results to the similar settings such as "8-bit + k6-bit" (kind of between 6bit and 7bit). Such baselines will show whether the proposed algorithm is really better than the default quantization methods with the same (or close enough) compression ratio.`
>
>     We agree that a more vivid comparison of methods is needed. For that reason we have conducted additional [experiments](https://github.com/quantizeit/kashin/blob/main/figures/glue.jpeg?raw=true)  on Glue, we quantized all layers of BERT/RoBERTa models, not falling back onto other methods and still produced superior results on this benchmark.  on GLUE benchmark, where we quantize linear layers in transformer blocks with several quantization methods with the same number of bits: uniform, kmeans, and Kashin quantization (ours) to ensure a fair comparison. As can be seen from the new results, Kashin quantization provides superior results with the same number of bits on almost all downstream tasks for the BERT model and all downstream tasks for roberta.
>
> Updated [version](https://github.com/quantizeit/kashin/blob/main/figures/paper.pdf?raw=true) of the manuscript considering the reviewers' comments is available on the anonymous [repo](https://github.com/quantizeit/kashin). We hope, that the provided additional experiments and details will be evidence for you to increase the overall score because we believe that the proposed approach could be useful for the research community and serve as a foundation for future research.

---

### Official Review · Reviewer_M4ym · 2024-03-22

**Q2-1 Originality-Novelty:** 3
**Q2-2 Correctness-Technical Quality:** 2
**Q2-5 Clarity Of Writing:** 3

**Q1 Summary And Contributions:**

The paper introduces the Kashin Quantization algorithm for data quantization based on Kashin representation principles. It decomposes vectors, matrices, and tensors into factors with small infinity norms, enabling efficient data compression. The algorithm maintains competitive model performance while achieving superior data compression, particularly in next-word prediction tasks using models like OPT. Theoretical properties and empirical evaluations highlight the algorithm's potential to transform data quantization practices in machine learning.

**Q2-3 Extent To Which Claims Are Supported By Evidence:**

3: Good: the main claims are supported by convincing evidence (in the form of adequate experimental evaluation, proofs, (pseudo-)code, references, assumptions).

**Q2-4 Reproducibility:**

2: Fair: key resources (e.g. proofs, code, data) are unavailable but key details (e.g. proof sketches, experimental setup) are sufficiently well-described for an expert to confidently reproduce the main results.

**Q3 Main Strengths:**

* Introduction of a novel Kashin Quantization algorithm for efficient data compression based on Kashin representation principles.

* Rigorous evaluation of the compression algorithm in next-word prediction tasks using models like OPT, demonstrating competitive model performance and superior data compression.

* Exploration of matrix decomposition techniques to reduce memory footprint and improve computational efficiency.

**Q4 Main Weakness:**

* Limited discussion on the practical implementation challenges of the Kashin Quantization algorithm in real-world large-scale language model applications.

* Limited exploration of the potential limitations or failure cases of the proposed Kashin Quantization algorithm in specific scenarios.

**Q5 Detailed Comments To The Authors:**

* Can you elaborate on the specific convergence properties of the algorithm and how the choice of basis vectors influences the convergence rate?

* What are the practical implications of using structured matrix types like Householder, DCT, and Butterfly matrices in the Kashin Quantization algorithm for memory efficiency and computation speed?

*  How does the algorithm address the challenge of quantizing activations in transformer models, especially in handling outliers originating from the attention mechanism?

**Q9 Complying With Reviewing Instructions:**

Yes

---

> ### Author Rebuttal · Authors · 2024-04-09
>
> Dear Reviewer, thanks for your feedback on the submission. We would like to carefully address the points you mentioned in weaknesses.
>
> * `Limited exploration of the potential limitations or failure cases of the proposed Kashin Quantization algorithm in specific scenarios. Can you elaborate on the specific convergence properties of the algorithm and how the choice of basis vectors influences the convergence rate?`
>
>     The convergence rate for the matrix kashin decomposition is close to linear(as shown on updated figure 3,with exception for householder/givens orthogonal matrices ), but the required number of iterations can be different. Apart from the choice of orthogonal matrix (we investigated its impact in Table 1 and Figure 3), we determined the type of the layer to be the most crucial factor. For example, q_proj layer converges in under 40 iterations and produces several distinct clusters of values while a fc1 layer converges in almost 4k iterations and results in bad clusterization. (figures 3 and 7). Note that both cases converge, meaning that the norm of difference X - U - QV tends to zero. However, it does not guarantee that values in  U and QV will form distinct clusters. The latter correlates with fast convergence. The “problematic” layers tend to be either output layers of the self-attention mechanism or fully-connected layers in the MLP block.  At the end of section 2 we theorised that assuming special (kronecker) structure of matrix Q, might impede some guarantees of the original algorithm, since in all the previous literature Q was assumed to be obtained from QR decomposition of a random matrix. Indeed in later experiments we tried to decompose the “problematic layers” with a vector algorithm applied to a flattened matrix and we get significantly better results for a random orthogonal than for a kronecker product of two smaller random orthogonals. It is still an open question as to why this occurs only for a smaller percentage of layers.
> * `Limited discussion on the practical implementation challenges of the Kashin Quantization algorithm in real-world large-scale language model applications. `
>
>     Kashin decomposition itself runs very fast (approximately 0.04 second for weights of OPT/Bert/Roberta models). But clusterization of U and QV takes up time. This means Kashin decomposition can still be used for quantization of weights and can be run in parallel but is not very applicable to dynamic quantization of activations. We are currently working on approximating cluster centroids from decomposition properties rather than kmeans algorithm. This will overcome the struggle with activations quantization.
> * `What are the practical implications of using structured matrix types like Householder, DCT, and Butterfly matrices in the Kashin Quantization algorithm for memory efficiency and computation speed?`
>
>     Kashin Quantization algorithm for matrices requires several multiplications by an orthogonal matrix on each iteration (Algorithm 2). As stated at the end of Section 3, structured orthogonal matrices allow for faster matrix-vector and, consequently, matrix-matrix multiplications. Specifically, instead of theoretical O(n^3) complexity for random orthogonal matrix, Butterfly and DCT matrices require O(n^2log(n)) for matrix-matrix multiplication. Householder reflection requires only O(n^2) multiplications. Storing a Householder matrix is equivalent to storing a single vector. DCT matrix can always be recalled through an explicit formula and butterfly and qr matrices - recovered from random seed. Furthermore, we can use the same Q for a set of weights with the same shapes (common case for LLMs, they usually consist of a large number of identical transformer blocks with repeating weight shapes).
> * `How does the algorithm address the challenge of quantizing activations in transformer models, especially in handling outliers originating from the attention mechanism?`
>
>     Indeed, activations of LLMs take up a lot of space and their quantization can be very beneficial for memory utilisation. However, quantization of activations is recalculated for each batch. For that reason the simplest and fastest methods (usually, uniform quantization) are used. Nevertheless, we are looking forward to try kashin quantization on activations, since it should be specifically efficient in handling large outliers, common for LLMs.
>
> Updated [version](https://github.com/quantizeit/kashin/blob/main/figures/paper.pdf?raw=true) of the manuscript considering the reviewers' comments is available on the anonymous [repo](https://github.com/quantizeit/kashin). We ask you to increase the score, because we believe that the proposed approach could be useful for the research community and serve as a foundation for future research.

---

### Official Review · Reviewer_nZK7 · 2024-03-23

**Q2-1 Originality-Novelty:** 3
**Q2-2 Correctness-Technical Quality:** 3
**Q2-5 Clarity Of Writing:** 4

**Q1 Summary And Contributions:**

The manuscript introduces a novel approach to quantizing LLMs based on the principles of Kashin representation to decompose data structures (vector matrices or tensors). The main idea of the decomposition is to break the tensor into two components with small infinity norms. This property leads to a more efficient representation, concentrating the entries of these factors around several peaks, which facilitates their replacement with fewer bits without significant loss of information or performance. The paper showcases the effectiveness of this method through experimentation on next-word prediction tasks using LLMs.

Main contributions include:
- Using Kashin representations for quantizing neural network weights
- A method for decomposing matrices without resorting to vectorization
- Theoretical and empirical analysis of the proposed method

**Q2-3 Extent To Which Claims Are Supported By Evidence:**

4: Excellent: all claims are supported by very convincing evidence (in the form of comprehensive experimental evaluation, rigorous mathematical proofs, detailed (pseudo-)code, precise references, well-motivated and realistic assumptions) and the authors deliver what they promise.

**Q2-4 Reproducibility:**

4: Excellent: key resources (e.g. proofs, code, data) are available and key details (e.g. proof sketches, experimental setup) are comprehensively described for competent researchers to confidently and easily reproduce the main results.

**Q3 Main Strengths:**

- Novel approach to data quantization: Kashin's algorithm for LLM quantization seems to pay off. Experiments show that this approach is particularly effective in decomposing data structures into components with small infinity norms, facilitating efficient quantization. The method maintains competitive model performance while significantly improving data compression rates.

- Memory efficiency: The proposed method uses structured orthogonal matrices (Householder, DCT, and Butterfly matrices) to avoid storing large, resource-intensive matrices. This approach benefits LLMs, which are known for their large memory requirements. These matrices not only enhance memory efficiency but also accelerate computation times, which in turn contributes to the overall effectiveness of Kashin Quantization.

- Comprehensive theoretical and empirical analysis: the paper analyzes the proposed algorithms' theoretical properties alongside convincing empirical evaluation. In Section 2, The paper suggests a connection between the convergence rate of the Vector Decomposition Kashin Algorithm and the Kolmogorov width, which is an interesting insight. Similarly, in Section 5, the manuscript shows encouraging results in terms of memory performance, such as "full 4-bit quantization significantly raises the model’s perplexity, whereas 4-bit + Kashin 6-bit quantization almost restores the quality of the 8-bit model".

**Q4 Main Weakness:**

- Choice of Orthogonal Matrices: It seems from the discussion in the manuscript that the efficiency of the Kashin Quantization relies on the choice of orthogonal matrices. While the paper explores various types of matrices (Householder, DCT, Butterfly), providing a more detailed discussion on the selection criteria for these matrices would be beneficial. What properties of these matrices make them suitable for the Kashin Quantization method? A more in-depth analysis of the role of these matrices in the quantization process would enhance the paper's clarity and relevance.

- Uneven quantization: In Section 4, the paper mentions that not all layers could be equally well compressed. This result could potentially lead to uneven efficiency and opens up the question of using hybrid approaches. Discussing how to address this issue or potential future research directions to mitigate this problem would be beneficial.

**Q5 Detailed Comments To The Authors:**

Details are given in the read strengths and weaknesses section.

**Q9 Complying With Reviewing Instructions:**

Yes

---

> ### Author Rebuttal · Authors · 2024-04-09
>
> Dear Reviewer, thanks for your feedback on the submission. We would like to carefully address the points you mentioned in weaknesses.
> * `While the paper explores various types of matrices (Householder, DCT, Butterfly), providing a more detailed discussion on the selection criteria for these matrices would be beneficial. What properties of these matrices make them suitable for the Kashin Quantization method? A more in-depth analysis of the role of these matrices in the quantization process would enhance the paper's clarity and relevance.`
>
>     Kashin Quantization algorithm for matrices requires several multiplications by an orthogonal matrix on each iteration (Algorithm 2). As stated at the end of Section 3, structured orthogonal matrices allow for faster matrix-vector and, consequently, matrix-matrix multiplications. Specifically, instead of theoretical $\mathcal{O}\left(n^3\right)$ complexity for random orthogonal matrix, Butterfly and DCT matrices require $\mathcal{O}\left(n^2log(n)\right)$ for matrix-matrix multiplication. Householder reflection requires only $\mathcal{O}\left(n^2\right)$ multiplications. Faster multiplication is beneficial not only for the application of Kashin decomposition but also for integrating the Q matrix into the model forward pass. As for convergence properties, we have demonstrated in Figure 2, and Figure 3  that Householder reflections perform poorly in comparison with dct, butterfly, and random orthogonal matrices both in vector and matrix cases. Furthermore, in section 3.5, we explained such behavior: low values of metric $\min_{\|x\|_2 = 1} \max( \Vert x \Vert_1, \Vert Q^Tx \Vert_1)$.
> * `Uneven quantization: In Section 4, the paper mentions that not all layers could be equally well compressed. This result could potentially lead to uneven efficiency and opens up the question of using hybrid approaches. Discussing how to address this issue or potential future research directions to mitigate this problem would be beneficial.`
>
>     Indeed, not all layers could be equally well quantized. Note that for all layers the algorithm eventually converges, meaning that the norm of difference $x - u - Qv$ tends to zero. However, it does not guarantee that values in $u$ and $Qv$ will form distinct clusters. The latter correlates with fast convergence. In cases of long convergence, the values are usually more uniformly distributed between min and max values which makes clusterization less accurate (Figure 7). The “problematic” layers tend to be either output layers of the self-attention mechanism or fully connected layers in the MLP block. At the end of section 2, we theorized that assuming a special (Kronecker) structure of matrix $Q$ might impede some guarantees of the original algorithm, since in all the previous literature $Q$ was assumed to be obtained from the QR decomposition of a random matrix. Indeed in later experiments, we tried to decompose the “problematic layers” with a vector algorithm applied to a flattened matrix and we got significantly better results for a random orthogonal than for a Kronecker product of two smaller random orthogonals. It is still an open question as to why this occurs only for a smaller percentage of layers. We are working towards tackling this issue. This will allow us not to use the hybrid approaches and utilize kashin decomposition quantization for all layers. Furthermore, in our latter [experiments](https://github.com/quantizeit/kashin/blob/main/figures/glue.jpeg?raw=true)  on Glue, we quantized all layers of BERT/RoBERTa models, not falling back onto other methods and still produced superior results on this benchmark.
>
> Updated [version](https://github.com/quantizeit/kashin/blob/main/figures/paper.pdf?raw=true) of the manuscript considering the reviewers' comments is available on the anonymous [repo](https://github.com/quantizeit/kashin). We hope, that the provided details and new promising experiments make it possible for you to increase the score because we believe that the proposed approach could be useful for the research community and serve as a foundation for future research.

---

### Official Review · Reviewer_QsBg · 2024-03-24

**Q2-1 Originality-Novelty:** 2
**Q2-2 Correctness-Technical Quality:** 3
**Q2-5 Clarity Of Writing:** 3

**Q10 Ethical Concerns:**

No.

**Q1 Summary And Contributions:**

The paper works on parameter quantization of LLM models and aims to reduce memory requirements and, when applicable, accelerate the computation. The essential contribution is a variant of the Kashin algorithm (representation) that effectively decomposes the original matrix/vector data into two vectors of magnitude-bounded entries through orthogonal matrices and greedy projection. The authors demonstrate the effectiveness of their methods by experimenting with the OPT model and QR/Butterfly matrices, achieving positive results in memory usage and perplexities.

**Q2-3 Extent To Which Claims Are Supported By Evidence:**

2: Fair: the main claims are somewhat supported by evidence (but the experimental evaluation may be weak, or does not match entirely with the claims, important baselines may be missing, proofs contain important ideas but lack rigor, algorithmic details are only discussed superficially, references are imprecise, assumptions are not sufficiently motivated or explicated, etc.).

**Q2-4 Reproducibility:**

2: Fair: key resources (e.g. proofs, code, data) are unavailable but key details (e.g. proof sketches, experimental setup) are sufficiently well-described for an expert to confidently reproduce the main results.

**Q3 Main Strengths:**

- The problem is well-motivated and has valuable practical implications. Through quantization, one can save memory and thus reduce operational costs for commercial applications that involve LLMs. In some cases, it is also necessary to ensure that the model size is well below the device capacity.
- The generalization from the vector Kashin algorithm to the matrix one seems natural. The proposed approach has a clear advantage over the naive approach and uses only square-rooted space for storing the orthogonal matrices.
- The experimental results look positive. In particular, 80% or more layers can be quantized from 8-bit to 6-bit precision with relatively minor performance degradation.
- The paper is easy to follow and utilizes multiple examples to illustrate the claims and definitions.

**Q4 Main Weakness:**

- The method is limited in novelty as the Kashin Quantization is already an established algorithm, and the proposed method (Alg.2) is its natural variant. The technique thus unsurprisingly inherits multiple helpful properties, such as quick mat-vec operation, low storage usage, and bounded vector norms. Similarly, the DCT and Butterfly matrices are also well-known orthogonal matrices.
- The method's applicability is unclear, exhibited by significant quantization errors and slow convergence. In particular, unlike Kashin Quantization, the paper does not provide a theoretical analysis or guarantee for the matrix variant.
- The experiments lack baselines and comparisons. It would be reasonable to compare the method with some baseline algorithms, e.g., 8-bit + Uniform6-bit, and some well-known ML model quantization or distillation techniques. Also, besides making claims, I suggest the authors provide stats around inference speed and memory requirements to justify the "significant reduction." Similar stats should be available for the baselines and alternatives.
- Neither code nor data is available, which is a drawback as the paper justifies the algorithm's usefulness through experiments, not theory.

**Q5 Detailed Comments To The Authors:**

Multiple constructive comments and feedback are already in the prior sections. Please take time to review and address them. The following suggestions may also be helpful.
- There's still some space for a more detailed literature review. It would be better if the authors could state the pros and cons of their method compared to the existing ones.
- I wonder if the proposed method for LLM activations is challenging to implement and evaluate. Positive results in such experiments are probably more impressive than the current ones.
- Could the matrix Kashin algorithm be guaranteed to be fast in convergence by imposing some mild assumptions on the inputs?
- If that's the case, please state that the first few sub-sections of the Method section are from/based on Kashin and Romskii's work.

**Q9 Complying With Reviewing Instructions:**

Yes

---

> ### Author Rebuttal · Authors · 2024-04-09
>
> Dear reviewer, thanks for highlighting the novelty of the proposed approach, its memory efficiency, and the comprehensiveness of the analysis. We would like to carefully address your comments and the weaknesses you mentioned in the review.
> * `The method is limited in novelty as the Kashin Quantization is already an established algorithm,...` We would like to highlight that investigation into using different efficient orthogonal matrices has not been present in the previous works. Works of Kashin, Kashin & Romskii, Lyubarski & Vershinin used random orthogonal matrices (obtained from QR decomposition of a random matrix). Kashin & Romskii explicitly prove the Kashin theorem for such matrices. We not only proposed the matrix generalization of the Kashin algorithm but also researched the possible applicability of different classes of structured orthogonal matrices and theorized on the metric that can indicate suitable classes of matrices (Table 1).
> * `Neither code nor data is available ...` Updated [version](https://github.com/quantizeit/kashin/blob/main/figures/paper.pdf?raw=true) of the manuscript considering the reviewers' comments is available on the anonymous [repo](https://github.com/quantization4all/kashin) along with the source code to plot all the figures from the paper.
> * `If that's the case, please state that the first few sub-sections of the Method section are from/based on Kashin and Romskii's work.`
> Indeed, due to inadvertence, we have only mentioned Kashin and Romskii's work in the introduction, but it should also be mentioned in the Methodology section. We will correct that.
> * `The method's applicability is unclear ...` For the matrix variant the convergence rate is still asymptotically linear(as shown in [updated figure 3](https://github.com/quantizeit/kashin/blob/main/figures/convergence.jpeg?raw=true), with an exception for householder/givens orthogonal matrices), but the required number of iterations can be different. Apart from householder/givens orthogonal matrices (we explained their insufficiency in Table 1), we determined the layer to be the most crucial factor. For example, the q_proj layer converges in under 40 iterations and produces several distinct clusters of values while a fc1 layer converges in almost 4k iterations and results in bad clusterization. (figures 3 and 7). Note that both cases converge, meaning that the norm of difference $x - u - Qv$ tends to zero. However, it does not guarantee that values in $u$ and $Qv$ will form distinct clusters. The latter correlates with fast convergence. In cases of long convergence, the values are usually more uniformly distributed between min and max values which makes clusterization less accurate. At the end of section 2, we theorized that assuming a special (Kronecker) structure of matrix $Q$ might impede some guarantees of the original algorithm, since in all the previous literature $Q$ was assumed to be obtained from the QR decomposition of a random matrix. Indeed, in later experiments, we tried to decompose the “problematic layers” with a vector algorithm applied to a flattened matrix and got significantly better results for a random orthogonal than for a Kronecker product of two smaller random orthogonals. It is still an open question as to why this occurs only for a smaller percentage of layers.
> * `I wonder if the proposed method for LLM activations is challenging to implement ...` Indeed, activations of LLMs take up a lot of space and their quantization can be very beneficial for memory utilization. However, the quantization of activations is recalculated for each batch. For that reason, the simplest and fastest methods (usually, uniform quantization) are used. Nevertheless, we are looking forward to trying kashin quantization on activations, since it should be specifically efficient in handling large outliers, common for LLMs. The task we have to tackle for that is clusterization of obtained $u$ and $Qv$, which takes significantly more time than Kashin decomposition itself. We are working towards calculating approximate cluster centers from decomposition properties rather than kmeans. Which will boost the pipeline significantly and make it applicable for dynamic quantization of activations.
> * `The experiments lack baselines and comparisons ...` We have added [experiments](https://github.com/quantizeit/kashin/blob/main/figures/glue.jpeg?raw=true) with two more models (BERT and RoBERTa) on downstream tasks (GLUE datasets) where we provide a comparison to uniform quantization and kmeans quantization.
> * `Also, besides making claims, ...` In the experiments section, we have described the quantization pipeline specifics: which layers (linear) we quantize in int4/int5/int6 and which we leave in fp16 (layer norms, embeddings). In section 3 we describe the complexity of multiplication for different choices of orthogonal matrices.
>
> We believe, that additional experiments and insights may allow you to increase the overall score.

---

### Official Review · Reviewer_3n49 · 2024-03-26

**Q2-1 Originality-Novelty:** 1
**Q2-2 Correctness-Technical Quality:** 3
**Q2-5 Clarity Of Writing:** 3

**Q1 Summary And Contributions:**

authors propose to quantize LLM by orthogonal matrix decomposition techniques
evaluation on 125M and 350M models

**Q2-3 Extent To Which Claims Are Supported By Evidence:**

2: Fair: the main claims are somewhat supported by evidence (but the experimental evaluation may be weak, or does not match entirely with the claims, important baselines may be missing, proofs contain important ideas but lack rigor, algorithmic details are only discussed superficially, references are imprecise, assumptions are not sufficiently motivated or explicated, etc.).

**Q2-4 Reproducibility:**

3: Good: key resources (e.g. proofs, code, data) are available and key details (e.g. proofs, experimental setup) are sufficiently well-described for competent researchers to confidently reproduce the main results.

**Q3 Main Strengths:**

- problem is very relevant
- idea to us orthogonal matrix decomposition makes some sense to me

**Q4 Main Weakness:**

- evaluation only on small models
- algorithm unstable and does not scale and run very long (please correct me if i am wrong)
- no comparison to other methods
- comparison metric is very restricted
- very small LLMs were used only
- loss of performance is actually significant

**Q5 Detailed Comments To The Authors:**

- IMO there is too much space (6 pages) dedicated to orthogonal matrix decomposition which is well known (papers on convergence exist)
- authors should put more emphasis on the effect of their proposal -> memory savings, throughput improvements, getting rid of DDP , training + inference speed
- experimental baselines are missing completely
- better metrics could be applied

**Q9 Complying With Reviewing Instructions:**

Yes

---

> ### Author Rebuttal · Authors · 2024-04-09
>
> Dear Reviewer, thanks for your feedback on the submission. Thanks for pointing out that the neural network (especially, LLM) quantization problem is very relevant. We would like to carefully address the points you mentioned in weaknesses.
> * `Evaluation only on small models`,`very small LLMs were used only`
>     We acknowledge the limitation of our evaluation scope due to the resource limitation. However, it is important to note that the evaluation of small models remains highly relevant. Moreover, language models are mostly transformers with a repetitive structure of transformer blocks. Thus, quantization methods can be straightforwardly scaled on larger instances. Especially when talking about quantization of weights: such quantization can be performed in parallel.
> * `Algorithm unstable and does not scale and run very long (please correct me if i am wrong`
>     Kashin decomposition algorithm guarantees linear convergence for any input. We introduced a matrix version of the Kashin decomposition algorithm. This adaptation significantly enhances the algorithm's efficiency, reducing memory requirements from quadratic to linear per input. However, it's crucial to note that while our modified algorithm demonstrates superior performance in many cases, we observed that its convergence rate could be [slower](https://github.com/quantizeit/kashin/blob/main/figures/convergence.jpeg?raw=true) for certain types of input weight matrices. At the end of section 2, we theorized that assuming a special (Kronecker) structure of matrix Q might impede some guarantees of the original algorithm, since in all the previous literature Q was assumed to be obtained from the QR decomposition of a random matrix. Indeed, in later experiments, we tried to decompose the “problematic inputs” with a vector algorithm applied to a flattened matrix and we got significantly better results for a random orthogonal than for a Kronecker product of two smaller random orthogonals. It is still an open question as to why this occurs only for a smaller percentage of layers. The observed variations in convergence rates for certain inputs are reported and represent a targeted area for ongoing research. We also added some explicit statements about the convergence of the algorithms in the paper. As for the runtime of Kashin decomposition, 30-40 iterations(approximate number of iterations, needed for convergence for most layers, as shown in Figure 3. b) take around 0.04 seconds.
> * `no comparison to other methods`,`experimental baselines are missing completely`
>     Our manuscript does include a comparison with established quantization approaches. As shown in Table 2, we have employed the quantized weights for OPT models, as baselines for our analysis. These baseline comparisons specifically reference models quantized to "8 bits" and "4 bits," providing a direct comparison between our approach and existing quantization methods (to our knowledge, weights from the huggingface hub were quantized using quantile estimations).
>     Additionally, we have conducted [experiments](https://github.com/quantizeit/kashin/blob/main/figures/glue.jpeg?raw=true) on GLUE benchmark with two other models: BERT and RoBERTa, and compared our approach to uniform and kmeans quantization methods.
>     Furthermore, recognizing the importance of a clear experimental framework for replicating and understanding our results, we have supplemented our manuscript with additional information about our experimental setup in the “Experiments” section.
> * `comparison metric is very restricted`,`better metrics could be applied`, `loss of performance is actually significant`
>     We utilized the perplexity metric common for the evaluation of CausalLM models. However, it has its drawbacks, i.e., a small perturbation in loss can cause significant changes in perplexity. We have included an experiment on the GLUE benchmark which allows us to demonstrate our method’s efficiency with other metrics: accuracy, F1-score, Pearson/Spearman correlation, and Matthews correlation.
> * `IMO there is too much space (6 pages) dedicated to orthogonal matrix decomposition which is well known (papers on convergence exist)`
>     An overview of the Kashin algorithm for vectors collectively occupies approximately 2.5 pages. The rest include introducing a new matrix version of the Kashin algorithm and convergence speed analysis for different choices of orthogonal projection and analysis of resulting value distributions of $u$ and $Qv$. These topics have not been present in the previous works and constitute the novelty of our paper.
>
> Updated [version](https://github.com/quantizeit/kashin/blob/main/figures/paper.pdf?raw=true) of the manuscript considering the reviewers' comments is available on the anonymous [repo](https://github.com/quantizeit/kashin). We kindly ask you to increase the score, because we believe that the proposed approach could be useful for the research community and serve as a foundation for future research.

---

### Meta-Review · Area_Chair_GZJv · 2024-04-15

Summary: The research paper focuses on parameter quantization techniques for large language models (LLMs), with the goal of reducing memory requirements and potentially accelerating computational performance when applicable. The key contribution is a novel variant of the Kashin representation algorithm, which efficiently decomposes the original matrix/vector data into two vectors with bounded magnitudes, achieved through orthogonal matrices and greedy projection methods. The authors validate the effectiveness of their proposed methods through experiments involving the OPT language model and QR/Butterfly matrix factorizations. The results demonstrate promising reductions in memory usage while maintaining reasonable perplexity scores, indicating the practical viability of their quantization approaches for optimizing LLM deployments.

Meta-review: Based on the initial reviews, the verdict is that the authors have crafted an above-the-bar paper that presents a compelling proposal backed by solid reasoning. There is one pending reject, but I believe the authors have handled these issues sufficiently. Based on the interaction between authors and reviewers, most concerns have been raised, and all the reviewers place the paper above the acceptance borderline. The authors have adequately answered and interacted with the authors' questions.